# Simulation Informed CAD for 3D Nanoprinting

**DOI:** 10.3390/mi11010008

**Published:** 2019-12-18

**Authors:** Jason D. Fowlkes, Robert Winkler, Eva Mutunga, Philip D. Rack, Harald Plank

**Affiliations:** 1Center for Nanophase Materials Sciences, Oak Ridge National Laboratory, Oak Ridge, TN 37830, USA; prack@utk.edu; 2Bredesen Center for Interdisciplinary Research, The University of Tennessee, Knoxville, TN 37996, USA; emutunga@vols.utk.edu; 3Materials Science and Engineering, The University of Tennessee, Knoxville, TN 37996, USA; 4Christian Doppler Laboratory for Direct-Write Fabrication of 3D Nano-Probes, Institute of Electron Microscopy and Nanoanalysis, Graz University of Technology, 8010 Graz, Austria; robert.winkler@felmi-zfe.at (R.W.); harald.plank@felmi-zfe.at (H.P.); 5Graz Centre for Electron Microscopy, 8010 Graz, Austria

**Keywords:** focused electron beam induced deposition, 3D nanoprinting, Additive nanomanufacturing

## Abstract

A promising 3D nanoprinting method, used to deposit nanoscale mesh style objects, is prone to *non-linear* distortions which limits the complexity and variety of deposit geometries. The method, focused electron beam-induced deposition (FEBID), uses a nanoscale electron probe for continuous dissociation of surface adsorbed precursor molecules which drives highly localized deposition. Three dimensional objects are deposited using a 2D digital scanning pattern—the digital beam speed controls deposition into the third, or out-of-plane dimension. Multiple computer-aided design (CAD) programs exist for FEBID mesh object definition but rely on the definition of nodes and interconnecting *linear* nanowires. Thus, a method is needed to prevent non-linear/bending nanowires for accurate geometric synthesis. An analytical model is derived based on simulation results, calibrated using real experiments, to ensure linear nanowire deposition to compensate for implicit beam heating that takes place during FEBID. The model subsequently compensates and informs the exposure file containing the pixel-by-pixel scanning instructions, ensuring nanowire linearity by appropriately adjusting the patterning beam speeds. The derivation of the model is presented, based on a critical mass balance revealed by simulations and the strategy used to integrate the physics-based analytical model into an existing 3D nanoprinting CAD program is overviewed.

## 1. Introduction

Computer simulations are commonly employed to complement nanoscale synthesis as in situ characterization is often not practical. Thus, time-dependent dynamics operative during synthesis are often unresolved and reconstruction of multiple ex situ characterization steps is necessary to try to infer time-varying progressions. Unfortunately, single experiment time-dependent behavior is only approximated using characterization results derived from multiple experiments, whether using (i) multiple process times, using the same sample, or (ii) using multiple samples with variable process times. Computer simulations can thus facilitate understanding the physical, chemical and temporal coordinates, supplementing experimental knowledge to predict deposition geometry.

A simulation strategy is presented here that moves beyond simple geometric predictions, toward the integration of compensation strategies to avoid defects for enhanced nanoscale deposition precision. Specifically, simulation results, calibrated against real experiments, are used to construct an analytical mathematical model for the purpose of informing computer-aided design (CAD) to prevent structural distortions during deposition. At its core, the problem is of the reaction-diffusion type and depends on multiple time-dependent physical and chemical processes. The derived analytical mathematical model presented here represents the simplest description of the nanoscale synthesis method while capturing the critical physics that governs deposition. This approach is demonstrated for a 3D nanoprinting technique.

Three dimensional nanoprinting using focused electron beam induced deposition (FEBID) [1] is one such nanoscale synthesis method with limited in situ analysis options [2,3]. In this method, a focused (nanoscale) electron probe is used to continuously dissociate precursor molecules adsorbed in the beam interaction region (BIR). In the deposition mode, a deposit accumulates over time under both continuous precursor flow and continuous beam irradiation. In the etching mode [4], the substrate is consumed. 

Secondary electrons (SE) dominate deposition. SEs are created in the near surface region by the primary electron beam and the emitted SE surface flux drives precursor dissociation/fragmentation [5]. These experiments are carried out in either a conventional scanning electron microscope (SEM) or a dual electron/ion microscope equipped with a precursor injection nozzle capable of delivering molecules continuously, and at a relatively high flux, directly to the BIR. Localized precursor delivery preserves the overall high vacuum in the specimen chamber required for stable microscope operation. 

Using FEBID in the 3D nanoprinting mode [6,7,8], mesh style objects are deposited using a network design of interconnected linear nanowires. The typical nanowire diameter spans the length scale of 25–100 nm, so in situ optical imaging techniques are not practical for FEBID. The electrical current flowing through the substrate, to ground, can be dynamically monitored during deposition [9,10], which provides information of the status of deposition without influencing the deposition process. Deposit electrical properties have even been dynamically improved in situ using continuous electrical analysis coupled with machine learning [11,12]. However, information regarding the structural progress of deposition is limited. Film thickness measurements have also been carried out in situ, but this method is only appropriate for semi-infinite thin film deposition [3]. The lack of in situ characterization methods for FEBID, at least in part, motivated the creation of the 3D-FEBID simulation used here [13,14]. 

The main purpose of the current Communication is to report an analytical mathematical model, derived from 3D-FEBID simulations that corrects for unwanted structural distortions in the final deposit. Computer simulation results are presented that (i) provide the basis for the choice of a distortion correction model, namely modulating the exposure beam speed, and (ii) the core physics used to construct the model. The initial results of the correction scheme are presented, which paves the way for a more comprehensive future Article. Specifically, the correction scheme is discussed, with an emphasis placed of the role that precursor surface concentration gradients play in causing deposit distortion, whereas heating-related effects have been revealed previously in Utke et al., Randolph et al., Mutunga et al. and Skoric et al.’s researches [15,16,17,18]: this paper leverages heavily results previously reported in Mutunga et al.’s work [17]. Further, the integration of the analytical model with the FEBID specific, CAD environment (3BiD) [19] will be the topic of a future article.

## 2. Materials and Methods 

Standard nanoprinting settings for 3D-FEBID were used in the simulations, as summarized in Table 1. The relatively high primary electron beam energy (30 keV: range: 0–30 keV) minimizes electron elastic scattering in the growing deposit, thereby maintaining spatial resolution while an appreciable secondary electron flux is emitted to induce deposition on a tractable time scale (1–10 s). The relatively low beam current setting (32 pA: range: pA–μA) provides the highest spatial resolution while enabling adequate precursor replenishment in the BIR to facilitate continuous deposition. Winkler et al. [20] explains that under these favorable conditions for spatial resolution, 3D-FEBID still operates in a precursor limited reaction regime. The MeCpPt^IV^Me_3_ precursor emulated here is limited to monolayer surface coverage. Prior to FEBID, the initial precursor surface coverage is 0.66 (C_eq_ = 1.85 molecules/nm^2^·s), which is set by both the local precursor pressure, is P = 0.5 mTorr and the substrate temperature is T_o_ = 294 K. 

Simulation results are also provided for a different equilibrium precursor surface coverage of 0.50 (C_eq_ = 1.4 molecules/nm^2^·s), a condition achieved by lowering the vapor pressure to 0.25 mTorr at T_o_ = 294 K. Lastly, controlled 3D-FEBID has been previously demonstrated over the primary electron beam energy range of 5–30 keV [20]. Thus, simulations were also executed at E_o_ = 5 keV to span the full range of 3D nanoprinting as some initial measure of model robustness. In this case, the beam current was 25 pA, the beam size was 14.9 nm, P = 0.5 mTorr and T_o_ = 294 K.

The simulation of 3D-FEBID, written and compiled in Matlab^®^ (Version R2017b, Mathworks, Natick, MA, USA), consists of three main rate equations that collectively capture the integral physics and chemistry of 3D-FEBID.
(1)$$\frac{\partial C}{\partial t}=\nabla \left(D\left(T\right)\nabla C\right)+\frac{\delta \Phi }{s_{p}}\left(s_{p}-C\right)-\frac{C}{\tau \left(T\right)}-\sigma i_{SE}^{\prime \prime }C$$
(2)$$0=q_{b}^{\prime \prime \prime }+k\nabla^{2}T$$
(3)$$\frac{\partial V}{\partial t}=\frac{\Omega }{s_{d}}\sigma i_{SE}^{\prime }C$$

Table 2 defines the independent and dependent variables appearing in Equations (1)–(3).

The rate equations that govern FEBID are described with numeric references to the terms appearing on the right-hand side of each equation, moving from left to right. The surface precursor concentration (*C*) evolves in (x,y,z) according to Equation (1) considering (i) precursor surface diffusion on the deposit, (ii) vapor phase precursor replenishment through surface adsorption, (ii) precursor surface desorption and (iv) SE-induced dissociation. Simultaneously during FEBID, the primary electron beam transfers energy to the deposit through inelastic electron energy loss, resulting in (i) Joule heating and subsequent (ii) conduction heat transfer to the substrate heat sink (Equation (2)). Finally, a physical deposit evolves in the simulation spatial domain in direct proportion to the quantity of precursor dissociated and this is described by rate Equation (3)—the FEBID deposition rate at the BIR is directly proportional to the precursor surface concentration in the BIR. 

A detailed description of the simulation is provided in Fowlkes et al. and Mutunga et al.’s researches [13,17]. Here, only specific aspects relevant to the current work are provided. The finite difference method is used to simultaneously solve Equations (1) and (2), although Equation (2) is only updated when the primary electron beam is displaced to a new exposure pixel or if the quantity of deposition (Equation (3)) exceeds roughly one monolayer since the last temperature profile *T*(x,y,z) update. The steady-state approximation for heat transport is possible, in part, because the heat transport time scale is 3 orders-of-magnitude faster than the monolayer deposition rate in the BIR [17]. Equation (3) is updated following each simulation time step to evolve the deposit volume. Lastly, the parameters *q_b_’’’* and *i_SE_* are derived from Monte Carlo simulations of the electron-solid interaction [17]. This calculation is also updated on the order of the monolayer deposition time.

A 3D-FEBID simulation of a ‘calibration structure’ [13] is provided in Figure 1a. Deposition of the full range of segment angles ζ = 0–90° makes it possible to build a wide variety of mesh style objects [21]. A calibration structure consists of a vertical ‘pillar’ supporting a suspended ‘segment’. The pillar exposure element is a vertical nanowire satisfying ζ = 90°. A segment element constitutes all other possible unique exposure element angles 0 ≤ ζ < 90°. 

The pillar exposure element is deposited using a stationary primary electron beam for an extended time period (τ_d_ = 3.258 s). The segment element is deposited using a constant primary electron digital beam speed (v_b_ = Λ/τ_d_ = 1 nm / 8.19 ms = 122 nm/s). Segment exposure begins at the pillar axis position with deposition proceeding off the pillar tip in response to lateral primary electron beam displacement in the (x,y) focal plane. 

Computer-aided design carried out using the 3BiD program [19] requires multiple calibration structure deposition experiments spanning the segment angle range of ζ = 0–90° (Appendix A); the (*ζ*) is measured for a set of beam exposure speeds, defined by a variable (*τ_d_*) at constant (Λ). A calibration list is generated {*ζ*, *τ_d_*}. Next, the program calculates the required (*ζ*) for each exposure element defining the mesh style object. The value of (*τ_d_*) required to deposit each exposure element is then interpolated from the {*ζ*, *τ_d_*} list. The deposition order of exposure elements is also defined using the program. 

Linear deposition elements are defined in the 3BiD program and real deposition ensues under the assumption that a linear nanowire will be deposited. Returning momentarily to the calibration, (*ζ*) is measured at a projected length (x’ = 250 nm) for each segment calibration element [20]. However, the simulation presented in Figure 1a demonstrates that linearity is achieved only over a limited range, typically (<500 nm). Figure 1a suggests that the precursor surface concentration, which varies along the length of the segment, or *C*(*s*), is a key feature leading to non-linear segment deposition because this value steadily decreases along the segment length and the deposit growth rate (Equation (3)) is directly proportional to (*C*). Results are now presented that link the observed segment distortion with the precursor surface concentration spatial profile *C*(*s*).

## 3. Results

The deposition of a typical calibration structure was simulated to predict the precursor surface concentration profile *C*(*s*) along the 3D deposit at various values of total deposit length (S_T_). These data are presented in Figure 2a for a total deposit length of S_T_ ≅ 650 (red), 890 (yellow), 1120 (green) and 1300 nm (blue), see solid lines. Importantly, the solid lines represent deposition without distortion compensation, see Figure 1a. The presentation of the results focuses on the segment deposition, which takes place after pillar deposition, i.e., S_T_ > 500 nm. The complementary temperature profile for each case is shown in Figure 2b. (*C*) and (*T*) represent averages derived from data acquisition nodes (x,y,z) spaced evenly along the centerline of the deposit. Only surface deposit voxels contribute to the average and to contribute to this average, they must lie within a spherical volume, centered on each node, with a characteristic radius of 40 nm. Moreover, each *C*(*s*) profile, shown in Figure 2a, was derived at the end of the pixel dwell time, just prior to the beam displacement to the next exposure pixel. Lastly, in the results analysis presented below, *C*(*s*) will be used to refer to any variation in (*C*) or (*T*) along the segment, at a fixed segment length (S_T_), e.g., each line profile shown in Figure 2. At other times, it is useful to discuss how the precursor surface concentration at the segment tip, or BIR, varies with total segment length, i.e., *C*(S_T_). For example, the four circular data points shown in Figure 2 represent the *C*(S_T_) trend. 

### 3.1. Precursor Surface Concentration Outside the Beam-impact Region

To begin, we will examine the surface concentration and temperature for a total deposit length of S_T_ ≅ 1300 nm (**-**). *C*(*s*) steadily decreases, with increasing *s*, over the range s = 0–1200 nm. This trend is directly caused by the thermal gradient directed along the deposit length (Figure 2b, **-**). Joule heating in the BIR is induced by the inelastic scattering of the internally scattered electrons as they pass through the deposit (Figure 1a). The magnitude of Joule heating scales with (i) the z-thickness of the segment in the BIR which controls the quantity of energy absorbed during beam transmission, (ii) the total length of the deposit which increases thermal resistance and promotes heating, and (iii) inversely with the cross-sectional area (*A*) of the segment as the thermal resistance is inversely proportional to this area [17]. Thermal resistance (*R_T_*) increases due to points 2 and 3, i.e., *R_T_* = S_T_/(k × *A*). The exact role that beam heating plays is revealed by way of the complementary *C*(*s*) profile (●**-**). This hypothetical *C*(*s*) profile represents the equilibrium precursor surface concentration (*C_eq_*) that would be present along the segment if it were possible to heat the segment with the beam at the segment tip without consuming the precursor. This is implemented by calculating (*C_eq_*) solely based on physisorption (terms 3 and 4 in Equation (2)) and *T*(*s*) (Appendix B). In other words, this value represents the maximum possible value of (*C*), at any position (*s*), given the current surface temperature of (*T*). The simulated profile *C*(*s*) nearly equals *C_eq_*(*s*) over the range s = 0–1200 nm indicating that temperature-dependent vapor phase replenishment/desorption determines *C*(*s*) outside the BIR. The limitations of physical adsorption model are briefly described in Appendix B.

### 3.2. Precursor Surface Concentration Inside the Beam-Impact Region 

A drastic reduction in precursor surface concentration occurs in the BIR (s > 1200 nm, Figure 2a, **-**) over the distance of ∆s = 80 nm measured from the end of the segment. The mechanism for the formation of this concentration gradient (d*C*/d*s*) has been reported previously (Figure 10, in [17])—SE induced precursor dissociation (term 4, Equation (1)) generates a precursor gradient at the BIR boundary that maintains precursor flow to the BIR by surface diffusion (term 1, Equation (1)). Unfortunately, the relatively large MeCpPt^IV^Me_3_ organometallic molecule has a small surface diffusion coefficient. How, then, does precursor surface diffusion sustain deposition? Precursor surface diffusion is sustained by the large concentration gradient self-imposed by beam dissociation in the BIR, i.e., precursor flux ~ *D* × d*C*/d*s*. The compensation strategy proposed later hinges on the stability of this physical process. The stability is now quantified using a normalization procedure, as outlined below. 

### 3.3. Segment Deposition—A Stable Physical Growth Mechanism

Multiple precursor surface concentration profiles *C*(*s*) are shown in Figure 2a as a function of the total deposit length (S_T_). The general function shape of *C*(*s*) is approximately constant, independent of S_T_. This is clearly revealed by normalization (Appendix C) of *C*(*s*) for each S_T_. First, the normalization *C*/*C_eq_* reveals that physical absorption controls precursor replenishment along the segment element surface, except at the segment tip region, i.e., *C*/*C_eq_* ≈ 1. Secondly, the relatively steep precursor gradient, observed at the BIR, is found to be independent of (S_T_). This is clearly revealed when the origin for *s* is shifted to the total deposit length (S_T_) (Appendix C). The normalization analysis indicates that the physical mechanism driving 3D deposition is stable during segment growth, over a critical range of useful 3D nanoprinting (S_T_ < 1000 nm). The important question now arises, “If the physical mechanism driving deposition is stable, what then causes the non-linear distortion?”. The answer is: the increase in the precursor surface desorption flux inside, and outside, the BIR which occurs in response to the steady temperature rise at the segment tip as (S_T_) increases (Figure 2b, ●→●). This causes a complementary reduction in *C*(S_T_) (Figure 2a, ●→●) and the final geometry exhibits a non-linear segment deflection (Figure 1a).

## 4. Discussion

### 4.1. A Compensation Model for the Case of Non-linear Segment Deposition

The 3D-FEBID simulation reveals that the variation of the precursor surface concentration in the beam impact region *C*(S_T_), due to *τ*(*T*), plays a pivotal role in determining segment linearity. The deflection compensation strategy aims to offset the continuous decrease in *C*(S_T_) using beam patterning deceleration. Specifically, a continuous beam deceleration is proposed, where the pixel point pitch (Λ) remains fixed, while the primary electron beam dwell time per pixel (*τ_d_*) is appropriately increased. Recently, a high-quality helix geometry was deposited using FEBID by experimentally tailoring beam deceleration [22]. Thus, the performance of the strategy relies on the quality of the prediction for (*τ_d_*) as a function of (S_T_).

Derivation of the mathematical model begins with the vertical growth rate in the segment reference frame (Figure 1c, {x’,z’}). The FEBID vertical deposition rate (d*z’*/*dt*) during linear segment deposition is defined as
(4)$$\frac{dz^{\prime }}{dt}=\Omega \sigma i_{SE}^{\prime \prime }C$$

In the ideal linear approximation of 3D-FEBID, *C* is not a function of segment length and consequently, not time dependent (Appendix D). Unfortunately, this is not the case for the simulations reported here and a spatial dependence must be introduced. First, a chain rule expansion introduces beam deceleration into the vertical growth rate expression;
(5)$$\frac{dz^{\prime }}{dx^{\prime }}\frac{dx^{\prime }}{dt}=\Omega \sigma i_{SE}^{\prime \prime }C\left(x^{\prime }\right)$$
Only then is the constraint of constant precursor surface concentration relaxed to *C*(*x’*). It is more convenient to execute the expansion in (*x’*), instead of (S_T_), because the beam patterning takes place in (*x’*); primary electron beam scanning deceleration in the focal plane is represented by (d*x’*/d*t*) which correctly traces the projected length of the segment in this plane as a function of time. 

The next mathematical step is to specify the desired segment angle. The term (d*z’*/d*x’*) is used to introduce (*ζ*); if a linear deposition mode is desired, then this value should remain constant during segment deposition. These facts validate the use of tan *ζ* = d*z’*/d*x’*. Equation (5) is now cast in a more useful form by (i) imposing the design, or CAD specification for (*ζ*), (ii) separating constant parameters (left-hand side) from variable parameters (right-hand side) and (iii) converting from beam deceleration (d*x’*/d*t’*) to electron dose per unit length (d*t’*/d*x’*):(6)$$\frac{dt}{dx^{\prime }}C\left(x^{\prime }\right)=\frac{\tan\zeta }{\Omega \sigma i_{SE}^{\prime \prime }}$$

Thus, with a proper description of the decrease in *C*(*x’*), one can easily compute the required increase in the quantity (d*t*/d*x’*) to ensure that the right-hand side remains constant during segment deposition (Appendix E). Fortunately, the simulation results presented above in concert with results reported in Mutunga et al.’s research [17], identify a pathway to determine *C*(*x’*).

### 4.2. The Definitive Surface Mass Balance Dictating 3D-FEBID

The 3D-FEBID simulation reveals a surface mass, i.e., precursor, flow rate boundary condition that primarily dictates the nanoprinting mode deposition physics. This boundary condition provides a means to determine the BIR precursor surface concentration as a function of projected segment length *C*(*x’*) (Appendix F). Specifically, segment deposition is strongly controlled by the balance between the precursor surface diffusion rate and the precursor dissociation rate at the BIR boundary, or;
(7)$$D\frac{dC}{ds}P_{\zeta }=\sigma i_{SE}C$$
where the units are molecules/s. Linearization of the boundary condition makes it possible to cast the equation in terms of known quantities, derived from simulation
(8)$$D\frac{C_{\mathrm{eq}}\left(T\left({x^{\prime }}_{b}\right)\right)-C_{\mathrm{BIR}}}{\Delta s}=\sigma i_{SE}C_{\mathrm{BIR}}$$
where *x’_b_* is the position of the BIR boundary. Several assumptions were made in constructing this simplification, which are based on the simulation results presented in the Results section. The d*C*/d*s* region was measured from Figure 2a as ∆s = 80 nm, which is stable as a function of segment length (Appendix C). The BIR boundary concentration is taken as the equilibrium concentration (*C_eq_*), determined by the balance between adsorption and desorption, since the agreement between *C*(*s*) and *C_eq_*(*s*) extends nearly to the BIR boundary during segment deposition (Figure 2a, **-** vs **●-**). 

### 4.3. The Dwell Time Compensation Model for Linear 3D Nanoprinting by FEBID

The analytical dwell time compensation model, so-called because the dwell time is the variable parameter in (x’), may now be constructed by combining Equations (6) and (8) and Appendix B;
(9)$$\frac{dt}{dx^{\prime }}\left(x^{\prime }\right)=\frac{\frac{\tan\zeta }{\sigma \Omega i_{{SE}^{\prime \prime }}}\left(1+\frac{\sigma i_{SE}\Delta s}{P_{\zeta }D\left(T\right)}\right)}{\frac{\delta \Phi }{\frac{\delta \Phi }{s_{d}}+\frac{1}{\tau \left(T\right)}}}$$

A complementary temperature prediction is required as a function of segment length;
(10)$$T\left(x^{\prime }\right)=T\left(0\right)+\frac{dT}{dx^{\prime }}x^{\prime }$$

A thermal circuit network model was integrated with the 3BiD program, based in part on the 1D thermal modelling reported in Mutunga et al.’s research [17]. The network model is an automated prediction and will be the topic of a future article. Please note that dwell time compensation has only been explored for a carbon-rich nanoparticle-matrix composite, e.g., PtC_5_, in the limit where the amorphous carbon phase dominates heat transport [17]. In fact, it is possible that metal-rich deposits will not require exposure correction for heating effects because the relatively larger thermal conductivity of metals.

Digital beam scanning is used during real 3D-FEBID experiments, whereas the analog function was derived for compensation (Equation (9)). The digital form can be derived from the analog model for any given electron beam position as
(11)$$\tau_{d}\left(n\cdot \Lambda \right)=\overset{n\cdot \Lambda }{\underset{\left(n-1\right)\cdot \Lambda }{\int }}\frac{dt}{dx^{\prime }}dx^{\prime }$$

The variable *n* is an integer and defines the beam displacement in integer multiples of the pixel point pitch (Λ). *n* begins at the initial value of n = n_o_, where n_o_ >> 1 which delays the compensation to allow for segment nucleation, and a final value of n = N which is the total number of scanning pixels required to deposit the complete segment exposure element.

The 3D-FEBID simulation with the applied correction is presented in Figure 1c. The complementary *C*(*s*) and *T*(*s*) profiles are presented in Figure 2a,b, respectively. For clarity, Figure 3a,b show 2D slices through the simulation domain for the calibration structure images presented in Figure 1a,c.

The dwell time compensation method works by steadily increasing the deposition time per pixel to compensate for the steady decline in the precursor surface concentration in the BIR as the segment length increases. Ultimately, this yields a constant vertical growth rate per pixel and linear growth. As expected, the compensation tactic is accompanied by a temperature increase, for any given segment length (see Figure 2b, **--** for all colors) because the z-thickness of the deposit is preserved along the segment length (Figure 3b). Conversely, during uncorrected deposition, the thickness steadily decreases (Figure 3a) due to the steadily decreasing growth rate, reducing the magnitude of Joule heating (Figure 3a)—Joule heating scales proportionally with z-thickness in the BIR. 

A refresh time tactic is an enticing alternative to the dwell time compensation method where the beam is periodically displaced away from the deposit to allow for periodic cooling. Unfortunately, beam heating is implicit and cannot be avoided practically because the time scale to achieve steady-state heat transfer is on the order of 10^−7^ s while the total deposition and refresh times per pixel are on the order of 10^−3^ s for 3D-FEBID. Thus, the beam would have to be blanked on the order of 10^4^ times per exposure pixel to deposit only at a low temperature. This exposure frequency approaches the instrumentation limit. Furthermore, deposition under these conditions would be time-dependent in precursor surface concentration (d*C*/d*t*) because the beam cooling/heating rate is dynamic at this frequency—operation under steady-state conditions is more predictable. Please note that it is not our intention to discredit a refresh time-based approach but rather to highlight the advantage of operating under steady-state conditions. Thus, although the dwell time compensation operates at a slightly higher deposition temperature during growth, steady-state deposition affords predictability.

Interestingly, the compensation equation anticipates the temperature-rise accurately enough to impose the correct linear compensation (Figure 3b) even though the temperatures reached during the corrected simulation exceed those achieved during uncorrected deposition. This is probably due to the stability of the precursor flow balance at the BIR boundary (Appendix C) that ultimately minimizes the segment non-linearity, making compensation possible. Thus, pixel dwell time compensation appears to work, not only in an interpolated way, but also seems to extrapolate correctly. Only a comprehensive experimental examination can confirm this speculation.

The dwell time compensation method was also demonstrated at a lower precursor surface concentration, where the precursor vapor pressure was reduced from P = 0.5 mTorr to P = 0.25 mTorr during deposition (Figure 4). Ultimately, the decrease in pressure translates into a general decrease in *C* at all points *s*, including the BIR. Figure 4b shows that the compensation method successfully drives linear segment deposition as expected—even though C_BIR_ does not explicitly appear in Equation (9), the term appears as a variable in the derivation.

The dwell time compensation method was also tested at the lower end of the 3D-FEBID primary electron beam energy range (E_o_ = 5 keV) to explore the utility of compensation over the available nanoprinting range specified in the Methods section (Figure 5). GIS conditions were held constant relative to the simulation results reported in Figure 3 for E_o_ = 30 keV for comparison purposes. The electron–solid interaction volume is strongly confined to the BIR at E_o_ = 5 keV, whereas at E_o_ = 30 keV, the beam is only weakly scattered in the BIR. As shown below, the segment element (Figure 5a) gently curves downward for the uncorrected case (see x’ > 600 nm) while the dwell time compensation method (Figure 5b) is successful at producing a linear segment element.

The deposition simulation shown in Figure 5 was extended to a multi-level beam exposure to investigate the compatibility of the method with a nanowire ‘kink’, as well as a segment exceeding 1 μm in length. The kink was introduced in the CAD by exposing a second segment with an orientation of ζ = 54° and a length of 800 nm but with an in-plane/substrate rotation of π relative to the underlying supporting segment. Such a kink is a common feature in mesh object 3D models.

The dwell time compensation method was also successful at producing a complete set of linear, interconnected nanowires for this more complex geometry, suggesting that the model may be compatible with the kink feature (Figure 6b). For example, the superimposed arrow aligned with the initial segment is oriented at ζ = 54°. The measurement arrow was then flipped in the horizontal image dimension and moved to the second segment to show the change in segment angle for the second segment, relative to the first segment. A change in segment angle of ∆ζ = −2° was observed for the second segment to ζ = 52°. This example is intended to estimate the potential accumulation of segment angle errors as additional 3D nanoprinting exposure levels are added to a 3D mesh style object model. 

The multi-layer exposure simulation demonstration suggests that the dwell time compensation method may be extended to (i) more complex segment interconnections and (ii) exposure elements exceeding 1 μm. Only future experiments will confirm these simulated predictions.

## 5. Conclusions

The dwell time compensation method, strategically presented in Figure 1b, is now summarized. The 3D-FEBID simulation was used to pinpoint the most relevant physics which controls segment deposition during 3D-FEBID. The 3D-FEBID simulation revealed that Joule heating, caused by the transmitted primary electron beam, induces a temperature-dependent precursor surface desorption flux gradient along the segment. Ultimately, this gradient induces a steady reduction in the precursor surface concentration in the beam impact region (BIR) as a function of segment length. Fortunately, the simulation also reveals that the precursor surface flow balance between surface diffusion and precursor dissociation is stable during segment growth. This stability at least minimizes the impact of the precursor surface concentration in the BIR, which is realized in the final segment geometry as a relatively slight non-linear deflection of the segment when a linear deposit is desired. 

Revelation of the 3D deposition mechanism made it possible to derive a simple analytical description of 3D-FEBID for direct application in the CAD phase of 3D-FEBID. Key parameters derived from the simulation, such as (i) the characteristic length of the precursor surface concentration gradient (∆s) feeding deposition in the BIR, (ii) the emitted secondary electron surface current and flux (*i_SE_* and *i_SE_’’*), and (iii) the segment temperature *T*(*x’*), are directly applied to the analytical model. This correction should, in principle, be valid for future deposition experiments at the primary electron acceleration energy and beam current settings for which they were derived. In the future, the dwell time compensation method will be integrated with the FEBID CAD 3BiD program for general use. The quality of the dwell time correction method is currently under study experimentally.

## Figures and Tables

**Figure 1 micromachines-11-00008-f001:**
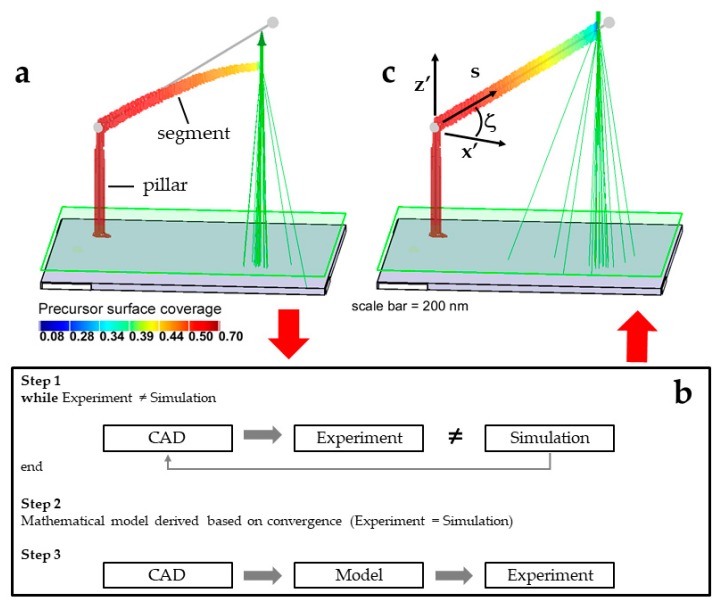
(**a**) A 3D-FEBID computer simulation of a so-called ‘calibration structure’ exhibiting a non-linear downward deflection/distortion causing the experimental final deposit to deviate from the CAD design (grey sphere-stick model). The precursor surface concentration is color-coded and presented as a fraction of monolayer coverage. The distance measured from the substrate along the deposit is given by (s), the total deposit length is (S_T_); (**b**) The proposed distortion correction scheme is based on (Step 1) a computer simulation that mimics experiments. In Step 2, the most important physics governing deposition is revealed which is incorporated into an analytical model. Ideally, the analytical mathematical model, called the ‘dwell time compensation method’, can be derived based on physical/chemical principles to ultimately correct for geometric distortions; (**c**) 3D-FEBID of the calibration structure following implementation of the analytical correction model described conceptually in the Discussion section.

**Figure 2 micromachines-11-00008-f002:**
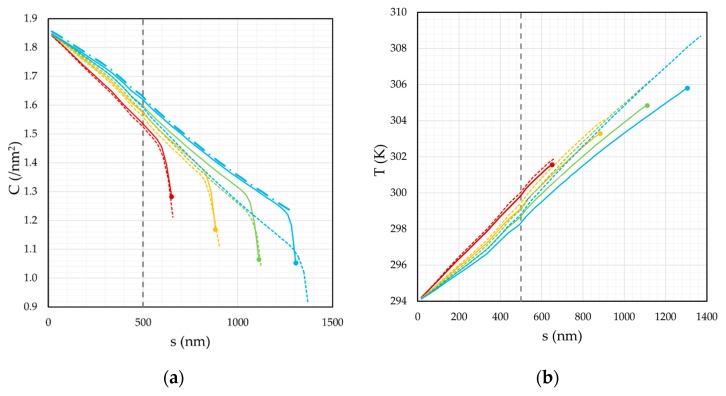
(**a**) The precursor surface concentration (*C*) along the deposit axis (*s*) at four different stages of deposition. The solid lines represent deposition without segment deflection compensation while the dashed lines represent deposition with compensation. Results are shown for a total deposit length (S_T_) ≅ 650 (**red**), 890 (**yellow**), 1120 (**green**) and 1300 nm (**blue**). The hatched grey line shows the boundary between the pillar (s < 500 nm) and segment (s > 500 nm) exposure elements.; (**b**) The complementary surface temperature (*T*) profile for the four stages of deposition shown in panel (a).

**Figure 3 micromachines-11-00008-f003:**
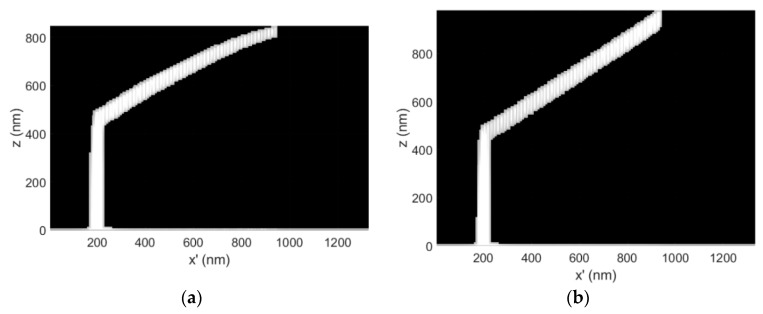
(**a**) 3D-FEBID simulation calibration structure cross-section for the uncorrected, as-deposited non-linear case. The dwell time per pixel τ_d_ = 8.19 ms for segment deposition, E_o_ = 30 keV, i_b_ = 32 pA and P = 0.5 mTorr; (**b**) The distortion corrected, complementary 3D-FEBID simulation cross-section demonstrating the deposition of the intended linear segment. In this case, the dwell time per pixel spanned the range τ_d_ = 8.19–12.23 ms. CAD specified a pillar length of 500 nm, a segment length of 1000 nm and a segment angle of ζ = 30°.

**Figure 4 micromachines-11-00008-f004:**
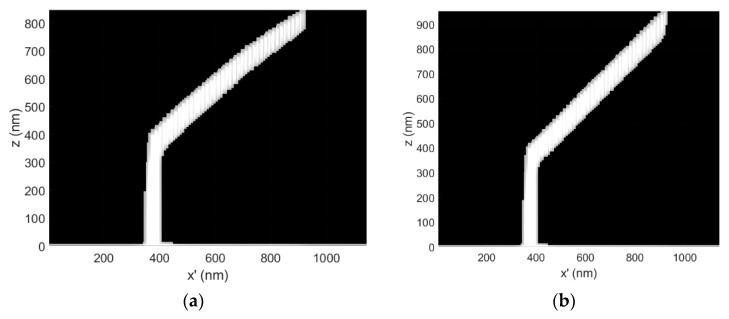
(**a**) A 3D-FEBID simulation demonstrating the uncorrected deposition of a calibration structure with a CAD specified segment angle of ζ = 54°, a pillar length of 400 nm, a segment length of 800 nm. The dwell time per pixel for segment deposition was τ_d_ = 15 ms, E_o_ = 30 keV, i_b_ = 32 pA and P = 0.25 mTorr; (**b**) The complementary 3D-FEBID simulation for the corrected situation. The dwell time per pixel spanned the range τ_d_ = 15–26.86 ms.

**Figure 5 micromachines-11-00008-f005:**
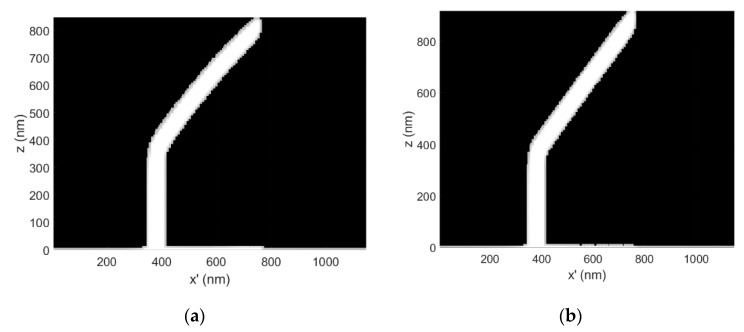
(**a**) A 3D-FEBID simulation of a calibration structure deposited at E_o_ = 5 keV and i_b_ = 25 pA. The CAD specified segment angle was ζ = 54°, the pillar element is 400 nm long and the segment element is 800 nm in length. The dwell time per pixel for the segment element was τ_d_ = 15 ms; (**b**) The distortion corrected, complementary 3D-FEBID simulation cross-section where the dwell time per pixel spanned the range τ_d_ = 15–21.94 ms during segment deposition.

**Figure 6 micromachines-11-00008-f006:**
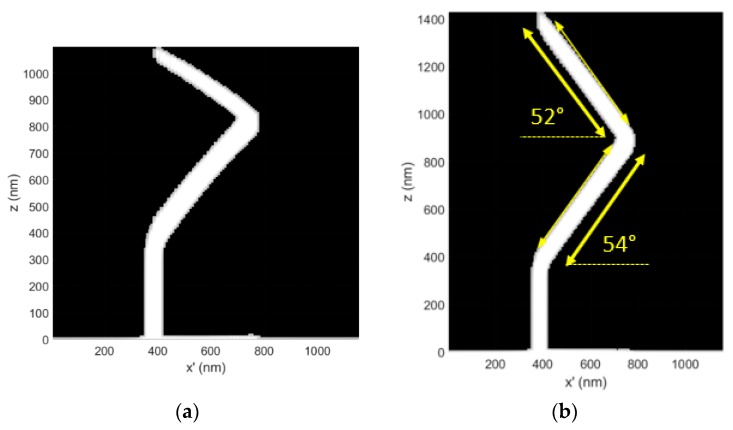
(**a**) A 3D-FEBID simulation of a multi-level deposit exposure without dwell time compensation using E_o_ = 5 keV, i_b_ = 25 pA, P = 0.5 mTorr and T_o_ = 294 K. The initial segment angle at take-off is ζ = 54° and the segment element length is 800 nm. A second segment element was deposited with an in-plane/substrate plane rotation of (π) to simultaneously simulate (i) a doubling of segment length and (ii) the incorporation of a ‘kink’ in the 3D geometry. Thus, the segment angle specified in the CAD file for the second segment is also ζ = 54°. (b) Dwell time compensation results in the desired linearization of the exposure elements. The dwell time range calculated for the first segment was τ_d_ = 15–21.94 ms. The second segment required the dwell time range of τ_d_ = 15–41.26 ms.

**Table 1 micromachines-11-00008-t001:** Simulation conditions for demonstration.

Parameter	Definition	Value
E_o_	Primary electron beam energy	5, 30 keV
i_b_	Primary electron beam current	25, 32 pA
	Primary electron beam size (FWHM)	14.9, 6 nm
	Precursor	MeCpPt^IV^Me_3_
τ_d_	Primary electron beam exposure dwell time	ms
Λ	Exposure pixel point pitch	1 nm
	Deposit composition	PtC_5_
	Substrate composition	5 nm SiO_2_/Si
T_o_	Substrate temperature	294 K
P	Precursor pressure at the substrate surface	0.25–0.5 mTorr

**Table 2 micromachines-11-00008-t002:** Focused electron beam-induced deposition parameters, definitions and units. Values for these parameters are reported in Table I [17].

Parameter	Definition	Units
*C*	Precursor surface concentration (x,y,z)	/m^2^
*T*	Temperature (x,y,z)	K
*V*	Deposit volume (x,y,z)	m^3^
*D(T)*	Precursor surface diffusion coefficient (x,y,z) via (*T*)	m^2^/s
*δ*	Precursor surface sticking probability	(0–1)
Φ	Precursor surface impingement flux	/m^2^·s
*s_p_*	Monolayer precursor surface coverage (0 ≤ *C* ≤ *s_p_*)	/m^2^
*τ(T)*	Mean precursor surface residence time (x,y,z) via (*T*)	s
*σ*	Mean, total electron impact dissociation cross-section	m^2^
*i_SE_*	Emitted secondary electron current (x,y,z)	e−/s
*i_SE_’’*	Emitted secondary electron flux (x,y,z)	e^−^/m^2^·s
*k*	Deposit thermal conductivity	W/m·K
*q_b_’’’*	Electron beam induced heating (x,y,z)	W/m^3^
Ω	Molecular volume of deposit for (PtC_5_)	m^3^
*s_d_*	Surface density of deposit (PtC_5_)	/m^2^
*P_ζ_*	Perimeter of pillar/segment nanowire	m

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
