# Peer review of "Simulation Informed CAD for 3D Nanoprinting"

_micromachines, 2019, doi:10.3390/mi11010008_

Round 1

Reviewer 1 Report

Fowlkes and co-workers outline an analytical/computational approach to improve the precision of CAD-guided FEBID of 3D nanostructures. By using Matlab simulations, the authors demonstrate a model than can effectively correct for non-linearity in FEBID processes, which is typically encountered in 3D FEBID experiments.
The results presented here are interesting and could lead to realization of significantly improved 3D FEBID nanostructures.

A few questions to the authors regarding the outlined modeling results:

As an example, the authors use a "pillar+segment" of a certain size i.e. height and lateral length. Are there limits in terms of size of deposited objects (height/length, diameter, ...) for which their model is valid? Could it be used to guide deposition of larger structures, say 5 micron objects?
If there is a size limit to the validity of the model, it would be helpful to the reader to state that clearly in the text.

In their model the authors use the value of 0.66 for surface coverage of precursor molecules. What results are expected if the surface coverage tends to be higher, closer to 1? In real experiments, for very high precursor supply fluxes, can one expect this value to exceed 1? Will that render the model invalid? Can you please comment on the validity of your model in terms of surface coverage of precursor molecules? How does a small variation in deposit composition (say from PtC5 to PtC6 or PtC4) affect the thermal gradient and what repercussions will that have on the validity of the model? In other words, how stiff is the model against the compositional changes of the deposited material? Is it only valid when the temperature varies negligibly over the BIR i.e. when the thermal gradient is very low,as mentioned in section 4.2?
To generalize this question, will this model be valid for other FEBID precursors, e.g. W, Au, Fe, Co... with the corresponding values of deposit thermal conductivity k?

A few technical remarks:

Line 100, equation (1): the last term (SE-induced dissociation) seems to be dimensionally incorrect. Can the authors please double-check this?

Similarly, line 102, equation (3): the term on the righthand side seems dimensionally incorrect. Can you please check this as well?
Perhaps in both instances one should be using emitted secondary electron flux Ise'' instead of emitted secondary electron current Ise, in line with notation given in table 2?
This should all be consistent with equation (7), which looks fine.

In table 2, some of the units seem to be written a bit oddly; to avoid confusion, one might use:
- precursor surface impingement flux in /m2s, instead of /m2/s
- emitted secondary electron flux in e-/m2s, instead of e-/m2/s
- deposit thermal conductivity in W/mK, instead of W/m/K
- surface density of deposit in /m2, instead of m2 (this one seems to be a typo?)

Similarly, in line 94 it would be better to write [molecules/m2s] than [molecules/m2/s]

The footnote at the bottom of table 2 seems to be "hanging in the air"; what part of the manuscript does it refer to, apart from pointing to ref 17? Perhaps this can be omitted althogheter.

Reviewer 2 Report

The communication entitled "Simulation Informed CAD for 3D Nanoprinting" from Fowlkes et al. expands on their previous works 15-17 and 19 to derive an analytical model of change of deposition rate due to temperature changes in the during the 3D-FEBID process to correct for structural non-linearities in the deposit. They show positive in-silico results, which they state they are trying to replicate experimentally in a future paper. The results are very promising for the development of accurate FEIBD processes to create complex micro/nano-machinery.

However, I am puzzled about one of the conclusions of the paper. One of the central premises of the article is that when the local temperature increases, the surface concentration of the precursor decreases. On p.7, line 234-236, they state that the concentration change can be compensated by increasing the dwell time. While I agree that the increased dwell time can compensate for a reduced growth rate, it would also increase the local temperature in the BIR even further, won't it? It seems that they also observed this temperature rise, as stated in p.9 paragraph starting in line 317.

I would have imagined if the temperature rise is the main problem, taking measures to eliminate temperature increase would be a more straightforward way. I can imagine that by decreasing the dwell time and including some wait time in between exposures to let the temperature equalize, one could stabilize the growth rate. Remember that the deposit is connected to a massive heat sink, i.e., the substrate, which should allow it to cool relatively quickly.

I might be getting something wrong here. Still, I would be happy if the authors could clarify why selecting parameters that even further increase the segment temperature is a better choice? I think an improved discussion on this issue in the abovementioned paragraph would also clear any questions that might pop up in the reader's mind.

In conclusion, I suggest publication of the paper in the journal Micromachines, after the points raised above are addressed.
